# Assessing Animal Welfare Risk in Fibre-Producing Animals by Applying the Five Domains Framework

**DOI:** 10.3390/ani13233696

**Published:** 2023-11-29

**Authors:** Kaja Salobir, Marlene Katharina Kirchner, Daniela Haager

**Affiliations:** FOUR PAWS International, Linke Wienzeile 236, 1150 Vienna, Austria; marlene.kirchner@vier-pfoten.org (M.K.K.); daniela.haager@vier-pfoten.org (D.H.)

**Keywords:** sheep, goat, cattle, waterfowl, alpaca, wool, cashmere, down, mohair, leather

## Abstract

**Simple Summary:**

There is an urgent need for a system that evaluates the welfare of the 5 billion farm animals involved in the textile industry. We aimed to create a risk assessment tool to evaluate the current state of the industry by assessing different textile standards that promise higher levels of animal welfare. Only 1 of 17 assessed standards reached an acceptable level of risk for animal welfare, with the rest falling behind. This shows that further work is needed when it comes to improving animal welfare within the textile industry, beyond depending only on the minimum requirements outlined by certifications.

**Abstract:**

Nearly 5 billion farm animals, including waterfowl, cattle, sheep, goats, and alpacas, are being affected by the fashion industry. There is an urgent need for a system that evaluates their welfare. The rise in public interest on the topic of animal welfare is leading to the creation of different textile standards or certification schemes, which can give us an overview of the general state of expectations in terms of animal welfare within the textile industry. We therefore created a risk assessment tool and applied it to 17 different textile standards. Our results showed that only one of the standards reached a score in the “Acceptable” animal welfare risk category, and the rest of the standards had even lower scores of risks for animal welfare. In general, industry standards have not demanded sufficient requirements for higher levels of animal welfare. While the current risk assessment gave us a good idea of what is considered acceptable within the industry, it is also not necessarily representative of the risks for the majority of farm animals that are part of the textile industry. Only a small number of animal-derived materials are certified with some form of animal welfare standards, even though these standards can play an important role in improving the standard of care for animals. To evaluate the actual welfare states of the animals in fibre production, further research is needed to apply the proposed tool to actual farms.

## 1. Introduction

Farm animal welfare and sustainable farming practices are becoming increasingly important to consumers. In a poll by YouGov from 2021, 86% of consumers agree that animal protection should be a company’s priority, along with upholding environmental and social standards [1]. Although the fashion industry is commonly considered to be a by-product of the meat industry, there are many farm animals that are affected by it either way, as the meat and textile industries are highly interlinked.

There are many different husbandry procedures and ethically questionable practices that farm animals, such as waterfowl, cattle, sheep, goats, and alpacas, are routinely subjected to. The most prominent are mutilations, such as castration without pain relief, tail docking, dehorning, mulesing, ear-notching, and wing clipping. Additionally, animals are exposed to inappropriate management practices, for example, early separation of young from their mothers, highly intensive concentrate feeding, and live feather plucking. Furthermore, intensive farming systems do not provide an appropriate physical environment for animals, and the animals are commonly kept isolated in cages, in crates, or tethered. This is then commonly followed by long-distance live animal transport, rough handling, and questionable slaughter practices. All these painful practices have a direct impact on the animals’ health, both physically and mentally [2]. The animals experience negative states of welfare throughout their lives and cannot fulfil their basic needs.

In the 2021 YouGov poll, 60% of the responders also believed that fashion brands should be responsible for animal welfare within their supply chain [1]. To give an incentive for brands to go well beyond just sparing the animals from certain painful practices, such as mulesing in sheep, and instead aim to ensure a good quality of life, we performed an in-depth investigation of a risk evaluation of 17 certifications for animal-derived materials that brands commonly use. We looked into the animal welfare requirements of each certification and how well those requirements held up against the Five Domains model for animal welfare [3].

The Five Domains model is the most up-to-date and comprehensive welfare assessment framework. It is an ethical framework that focuses on the internal and external conditions that give rise to an animal’s mental experiences. The effects of these various mental experiences represent the welfare status of an animal at a given time. A positive animal welfare state, based on the Five Domains model, describes the effect of all mental experiences of an individual animal at a given time and is characterised by the minimisation of negative experiences while also enabling positive experiences [3].

### Animals in the Textile Industry

Nearly 5 billion farm animals are affected by the fashion industry annually (Table 1) in some way or another. It is estimated that the global down and feather production volume was more than 500 thousand tons in 2022 [4]. More than 650 million geese and nearly 3 billion ducks are raised and slaughtered globally on industrial farms each year, with China being the leading production country in the world [5]. The exact number of animals used in leather production is highly difficult to define due to the nearly non-existent traceability of the supply chain. It nonetheless affects approximately 700 million cows, pigs, and small ruminants worldwide [6]. Approximately 1.2 billion sheep exist worldwide (FAO, 2020) [7], with approximately half of them directly used in the textile industry in Australia, China, and New Zealand [8,9]. More than 30 million cashmere and mohair goats [10,11] are used for the production of approximately 20,000 tons of cashmere [12] and 5000 tons of mohair [13] each year in China and South Africa, respectively, and more than 4500 tons of alpaca wool are produced annually from a population of more than 4 million Peruvian alpacas [14,15]. Additionally, there were also approximately 50 million Angora rabbits that were used to produce approximately 10,000 tons of Angora wool in 2002 [16]. This number has since dropped significantly, but it is nearly impossible to find more current data [17].

Despite the high numbers of animals that end up within the fashion supply chain, only a minuscule number of animal-derived materials are certified to some form of an animal welfare standard [4], even though they can play an important role in improving levels of traceability, as well as the standard of care for the animals. These certifications or standards can, however, give us an overview of the general state of expectations in terms of animal welfare within the textile industry. Currently, most of the animals used in the textile industry are kept in intensive husbandry systems, be it in cages, factory farms, or feedlots, with the exception of alpacas, which are kept mostly on nomadic, small-scale farms. The husbandry conditions of animals also differ among production countries, both due to the cultural differences in the keeping of animals, as well as due to different legislative requirements of different countries. More specifically, for example, cage keeping for waterfowl is not highly common within the European Union, while it is quite common in China—the main production location for down. Many such differences exist, especially in terms of different mutilations of animals; however, the animal needs are the same throughout the world (e.g., in the case of waterfowl, having water access for bathing, or appropriate feed) [18].

Due to the high number of animals that are directly and indirectly involved in the textile industry, we believe there is an urgent need for a reliable, science-based system that evaluates their welfare. To our knowledge, currently, no comprehensive welfare assessment data of animals in fibre production are available, also due to a lack of an evaluation tool that can be used for such a purpose. The aim of this study was therefore to create a risk assessment tool for farm animals in the textile industry, apply it to different standards, and then interpret the results of the initial risk assessment. We aimed to assess 17 major animal welfare certifications and other frameworks (animal welfare benchmarks, standards, or guidelines that do not necessarily lead to certification) for key animal-based textiles used by the fashion industry: alpaca, cashmere, down, leather, mohair, and wool. These certifications were all publicly available and well known within the textile industry.

## 2. Materials and Methods

Our materials and methods are separated into three parts, as per the aims of our study. First was the creation of the risk assessment tool, and second was the application in the form of assessing different textile certifications. Finally, we also defined the welfare evaluation of the overall score in terms of the actual risk that the animals have for respective welfare states, farmed under the minimum requirements of the respective textile standard.

### 2.1. Development of the Risk Assessment Tool

#### 2.1.1. Defining the Structure

To create the risk assessment tool, we first looked at all available welfare assessment protocols for farm animals and their structure [19,20,21,22,23,24]. The research was two-fold, with one focus on different available ethical frameworks used to define animal welfare as well as identifying a structure that can be used for a future score aggregation, such as the Five Freedoms [25], the Five Domains model [26,27], and the Welfare Quality^®^ (WQ) principles [28,29,30], considering we wanted to have one overall score.

We then chose the Five Domains model due to it being the most recent and updated version of defining welfare states from an animal’s point of view, as well as for its inclusion of mental states. According to this, a positive animal welfare state can be achieved by fulfilling five general welfare aims [26]:Good nutrition;Good health;Good physical environment;Appropriate behavioural interactions;Positive mental experiences.

These welfare aims can be translated into the Five Domains model and are set to minimise the negative internal states with the help of corresponding provisions (Table 2). The five general welfare aims were also what we took as the final structure into which we assorted our indicators.

#### 2.1.2. Defining the Indicators

Our second research focus for designing the tool was on exploring different indicators and parameters to define our research questions. The resources used for this were the international literature and science on animal welfare states as well as information on current farming practices and husbandry systems that are known to cause pain, harm, distress, and suffering to the animals [31,32,33]. The species we investigated for the risk assessment development were goats, sheep, alpacas, waterfowl, and cattle. Depending on the species (but generally aligned among them), we investigated what kind of requirements meet the basic welfare needs of an animal, for each general welfare aim.

We first defined species-specific single measures that were adapted from the WQ [28,29,34] and AWIN [21,22] animal-based parameters (e.g., “Body Condition Score”). Where no reliable animal-based parameters were available, we chose resource- and management-based indicators that would fit the respective animal production system. We searched for indicators until we covered all provisions with at least two single indicators for each species (Table 2).

Within the nutrition domain, we defined two provisions, namely appropriate feed and water. When it came to evaluating the aims of appropriate feed provision, we considered ad libitum access to roughage [35,36,37] and appropriate quantity of (additional) feed to the animals, as well as the regular measurement of Body Condition Score [38,39]. Appropriate water provision was evaluated by looking into the quality, unlimited quantity, and unrestricted access to water [40].

The physical environment domain was defined by three provisions. The first was environmental comfort, where we wanted to know about the proper shelter, which should have appropriate climate conditions (temperature [41,42,43,44] and air ventilation [45]) and protect animals from loud noises [46,47]. The second provision was resting comfort, where we looked into appropriate bedding and flooring [48], as well as providing an appropriate resting area [49,50,51]. The third provision within the domain was ease of movement, considering if animals are tethered [52] or kept individually, as well as looking into the minimum space requirements that animals are provided with, as overcrowding can be a serious issue for animal health [42,53,54].

The health domain had four provisions—absence of injuries, absence of disease, mutilations, and fitness. We evaluated the absence of injuries as daily injury checks within the herd and proper management of lameness [55,56]. The absence of disease was similarly measured by regular veterinary checks, preventative measures (parasites control, vaccinations) that are properly implemented, and appropriate healthcare—for example, by requiring that there are sick bays available on site. Within the mutilations provision, we investigated routine mutilations and the requirements regarding them [57,58,59,60]. While the number of mutilations differed slightly among the species (Table 3), at the criterion level, the worst score among the two/three/four partial scores (e.g., one for dehorning and one for tail docking) was retained. Allowing any other mutilations (e.g., ear-notching, biopsies of breeding animals, teeth clipping) immediately resulted in 0 points in the provision. The final provision in the health domain was fitness, where we were interested in whether the animals have year-round outdoor access (i.e., an outdoor run) as well as access to pasture [49,61,62,63].

The fourth domain was behavioural interactions, with three provisions. With the environmental interactions provision, we looked into if the animals are required to be provided with any enrichment in the environment [64,65,66], if their living areas are to be appropriately structured (e.g., in the case of goats, if they are provided with climbing possibilities [67,68]), and if there are any measures in place for improving comfort behaviour (e.g., scratch posts [69]). The social interactions provision investigated whether the animals are required to be kept in stable groups [70], if the young are allowed to be separated from their mothers [71,72], and if play behaviour is promoted in any way (e.g., keeping of animals in age-appropriate groups) [73]. The human–animal relationship (HAR) provision was focused on requiring positive handling of animals [74,75,76], not only during routine interactions but also during shearing. We also investigated whether there are any possibilities for the building of trust (e.g., by early handling of animals, regular contact without direct handling) and if it is required to habituate animals to routine husbandry procedures (e.g., shearing and combing, but also with veterinary checks).

The fifth domain, mental state, did not need any single indicators as the framework defines it as an interplay between the other four domains [3].

#### 2.1.3. Defining the Aggregation Procedure

After defining all indicators at the single-measure level, we developed the aggregation procedure, according to the hierarchy of the Five Domains model (Table 2). As we wanted to have one overall score for animal welfare risk, we aggregated scores from a single-measure level into a score per provision, by using a decision tree for each provision. These decision trees allowed us to weigh different single measures and their importance for their animal welfare state within the provision. The decision trees were led by the examples of weightings by the criteria level of Welfare Quality protocols on cattle [28] and poultry [23] (as the two closest species available), and relied on the comprehensive expert opinion and citizen involvement in this project [77], as well as on known literature on the topic (e.g., the mutilations decision tree was taken from WQ protocol [28] and adapted according to the species). A complete breakdown of the animal welfare risk evaluation measures in our review, in the form of decision tree questions, can be found in Appendix A, and their corresponding decision tree scores can be found in Appendix B.

Those provision scores were further aggregated into a score per domain, with the use of a Choquet integral to account for different weights the provisions carry within a domain. The Choquet integral drops the average to a lower score, according to the weights that have been predetermined with the use of experts’ opinion (see Appendix B for weight calculations and Appendix C for experts’ opinion) [78,79]. After that aggregation, we had four domain scores that were combined using a median for a final score per standard, with the underlying assumption that all four domains have an equal ability to influence the mental state of an animal [3] (see Table 4 for the aggregation flowchart).

### 2.2. Application of the Risk Assessment Tool

The certifications that we chose to evaluate covered six different animal-derived materials (alpaca wool, down, sheep wool, mohair, cashmere, leather), were available to the general public, and were known within the textile industry. We evaluated them between December 2021 and March 2022. This meant we carried out a comprehensive review of the 17 most popular and well-known certifications for animal-derived materials. We assessed the requirements of each certification that were publicly available and therefore clearly communicated, but we only scored for those that were obligatory and did not consider any possible recommendations from the standard or additional user manuals if they were not explicitly mentioned as a requirement.

We then looked at individual standard requirements and used them to answer questions regarding our single measures. The single measures (e.g., Body Condition Score) were translated into questions of whether such measures have been taken into consideration within the animal welfare requirements of the standard (e.g., “Is the Body Condition Score of the animals regularly monitored and are the workers knowledgeable in assessing it?”). These questions were answered in a yes–no format, and the decision trees of the single measures gave us a score per provision (see Appendix A for a full list of decision tree questions). Those were aggregated with the use of a Choquet integral into a score per domain and then finally combined into a single score for each individual standard.

### 2.3. Interpretation of the Overall Scores in Terms of Animal Welfare Risk

The evaluation part with the input of standards resulted in each certification receiving a single score that put the certifications into different categories of potential animal welfare risks (Table 5). The potential for animal welfare risk categories were adopted from WQ protocols [28]; however, one additional level of possible welfare risk was included to distinguish between risks for very poor and poor animal welfare.

Probability for very poor animal welfare (0–19) means that the risk of painful and ethically questionable practices and of animals suffering is very high. In most countries with animal welfare acts in place, it would probably be a violation of the respective legislation. This category requires immediate action to save animal lives and/or end prolonged suffering. Probability for poor animal welfare (20–39) means that the risk of most painful and ethically questionable practices is high and that we would expect prompt short-term improvements, along with substantial changes to counteract poor welfare states of the animals. Probability for acceptable animal welfare category (40–59) means that the risk of poor animal welfare states is present, but not very high. There are still improvements needed for the short or medium term in some areas, while some areas are not particularly harming the welfare states of animals. Probability for good animal welfare (60–79) means that the risk of poor welfare states is low, and only minor amendments are necessary, most likely in one or two areas only. Probability for excellent animal welfare (80–100) means that the risk of poor welfare states is very low, and animals most likely live a life worth living, encountering positive experiences and minimal negative experiences.

## 3. Results

The tool-creation part resulted in the creation of Appendix A, a breakdown of the animal welfare risk evaluation measures in our review, and its corresponding decision trees (Appendix B). The decision trees had a species-specific approach, with yes–no questions that were ranked and scored according to expert opinion, in the score range of 0–100. An example of a decision tree, displaying the appropriate feed provision, can be seen in Table 6, and the full collection is available in Appendix B, Table A2, Table A3, Table A4, Table A5, Table A6, Table A7, Table A8, Table A9, Table A10, Table A11, Table A12, Table A13, Table A14, Table A15, Table A16, Table A17 and Table A18.

To aggregate the results from a decision tree into scores per domain, we needed to define weightings for each provision with the use of expert opinion. This resulted in the creation of Appendix D: Choquet integral calculations (example of a Choquet integral calculation for the four domains and weight capacities and their interactions).

Finally, the assessment part resulted in 17 individual scores for each of the textile standards. Even though our evaluation was based on publicly available data, we believe sharing the individual results, along with the standard’s name, would not be the best course of action at this time; therefore, we have anonymised them (Table 7). Each certification therefore received a single score that was then interpreted into what it means in terms of an animal welfare risk. We turned the results of each standard into a colour-coded graphical representation, as seen in Figure 1 below.

Single measure criteria (i.e., the adapted animal-based parameters, “Body Condition Score measuring”) can be seen in the outermost circle. With the use of a decision tree, those measures gave us a score per provision (e.g., “appropriate feed”), and were then aggregated to a score per domain (e.g., “Nutrition”) with the use of a Choquet integral. Lastly, the four domain scores were further accumulated into a final, mental state, score as a median (innermost circle). This example has two domains that fall under the “Probability for very poor animal welfare” category, namely health and behavioural interactions. Nutrition and physical environment are, however, ranked as “Probability for poor animal welfare”, and the final score is therefore just above the threshold, in the “Probability for poor animal welfare” risk.

## 4. Discussion

To our knowledge, there are no currently available risk assessments for farm animal welfare, although more and more research on the topic has become available in recent years [80]. Due to our interest in animals in the textile industry, we wanted to create a risk assessment to evaluate the animal welfare of these farm animals and created a scoring system that includes the Five Domains framework [3,26,27]. We chose the Five Domains framework as it can be used as a basis for risk assessment of animal welfare states (and can also form a basis for future on-farm assessments). The Five Domains model currently does not offer a calculation or a scoring system—although the group’s recent article does refer to the possibility of using it as a risk assessment tool within the food industry standards and guidelines [81].

The previously well-known Five Freedoms [82] were a huge step towards improving animal welfare, but now they are outdated scientifically; they focus only on preventing animals from having negative experiences, disregarding their role in keeping animals alive and the existence of positive effects and their role in enhancing welfare [25,26]. Only by considering both can we hope to achieve realistic assessments of welfare states, and we do so by examining the interplay between the four domains—nutrition, physical environment, health, and behavioural interactions—and their effect on the fifth domain—mental state.

Our risk assessment tool was created by researching different existing protocols and their structure [19,20,21,22,23,24]. While these protocols are extremely comprehensible, they either do not offer the option of obtaining a single score from the assessment or are not really based on the Five Domains model and allowing for animals to be assessed on having positive experiences, both of which were our prerequisites. We then needed to define our single measures, based on the known animal welfare literature, as well as their importance within a provision, and create a decision tree, which was following the idea of Welfare Quality^®^ protocols [28] for, e.g., giving a score for mutilations and the absence of prolonged thirst. The decision tree scores (i.e., provision scores) were then aggregated with the use of a Choquet integral to the domain level to prevent compensation of a higher score in the midst. To calculate the integral, we needed to determine the weights of provisions within a corresponding domain. We did so with an exercise of experts’ opinion to calculate weights for Choquet integral and rank the importance of provisions within the domains (see Appendix C and Appendix D for more details), again following the same principle as the Welfare Quality^®^ protocols [29] and validated also in further literature [31]. The measurement of the fifth domain, the mental state, was defined as an interplay between the other four domains and was therefore calculated as a median, due to the underlying assumption that all four domains have an equal ability to influence the mental state of an animal [3].

There is a high number of farm animals involved in the textile industry, while simultaneously, there is a rise in public interest in the topic of animal welfare, leading to the creation of different textile standards that supposedly guarantee it. Certifications allow producers who invest in sustainable practices to be rewarded by the market. They also support brands in verifying the claims made to their customers, and customers rely on certifications to guide them in making choices that most reflect their values. Despite that, only a very small proportion of all animal-derived materials are certified to some animal welfare standards. For example, less than 3% of the world’s wool supply and just over 4% of the world’s down and feather supply are certified [4,9]. However, as our findings show, further work is needed when it comes to improving animal welfare within the textile industry. We wanted to see how far the certifications go in terms of their animal welfare requirements, and therefore, we created a risk assessment for potential welfare states of different certifications and gained a better understanding of the state of the industry.

Our results showed that only one of the standards we reviewed reached a score in the “Acceptable” animal welfare risk category, while the rest fell into the “Poor” and “Very poor” animal welfare risk categories. In general, standards have not prohibited most of the painful and questionable practices of fibre animal husbandry and have not demanded sufficient requirements regarding the animals’ physical environment and behavioural interactions, while also lagging in nutrition and health. We expected a number of standards to prohibit mutilations in animals, and none of them prohibit all routine mutilations, with castration (with or without appropriate pain relief) being the main one. Unsurprisingly, however, most of the wool standards we evaluated do prohibit mulesing in sheep, which is most likely due to different public campaigns [83,84,85,86] against it. There is also a common denominator among standards in that they rarely require ad libitum roughage, which is essential for all ruminants that are prevalent in the textile industry. Additionally, the behavioural interactions domain is highly neglected amongst a majority of standards, in some way or another. Providing environmental enrichment and ensuring the animals are experiencing appropriate social interactions with their species is not something that is common amongst the standards, nor is ensuring higher levels of human–animal relationships. This shows that further work is needed when it comes to improving animal welfare within the textile industry, especially if it depends only on the minimum requirements outlined by the certifications.

However, while the risk assessment gave us a good idea of seeing what is acceptable within the industry, it is also important to keep in mind that it is not representative of the risks of the majority of farm animals that are part of the textile industry. In general, less than 5% of all animal-derived materials are certified [4]; therefore, the reality for the rest might be completely different. While we are aware that many farmers may go well beyond the bare minimum that is required from them (either by a textile certification or by local legislation), this is not always the case, nor was it the objective of this project. We do, however, encourage farmers and workers of all types of husbandry systems to use this risk analysis on their farms and determine if they are at risk of poor animal welfare. All types of husbandry systems can be analysed with this assessment, from highly intensive farms to small-scale, hobby farms.

## 5. Conclusions

To evaluate the actual welfare states of the animals in fibre production, further research is certainly needed to apply well-designed and valid animal welfare assessment protocols to actual farms. Therefore, our next step will be to perform actual farm welfare assessments, by translating our work into on-farm evaluations. It would also be interesting to compare the results of the risk assessment with an on-farm evaluation that has been certified to a corresponding standard; however, we are aware that for those results to have any bearing, a very high number of on-farm evaluations would be needed.

With this tool and its results, we hope to communicate with certification owners regarding possible improvements in the standards, as we believe that responsibility and transparency along the supply chain remain an important part of the industry. By working together, animal protection organisations, fashion brands, standard owners, and producers can ensure animal welfare certifications develop a path of continual improvement.

## Figures and Tables

**Figure 1 animals-13-03696-f001:**
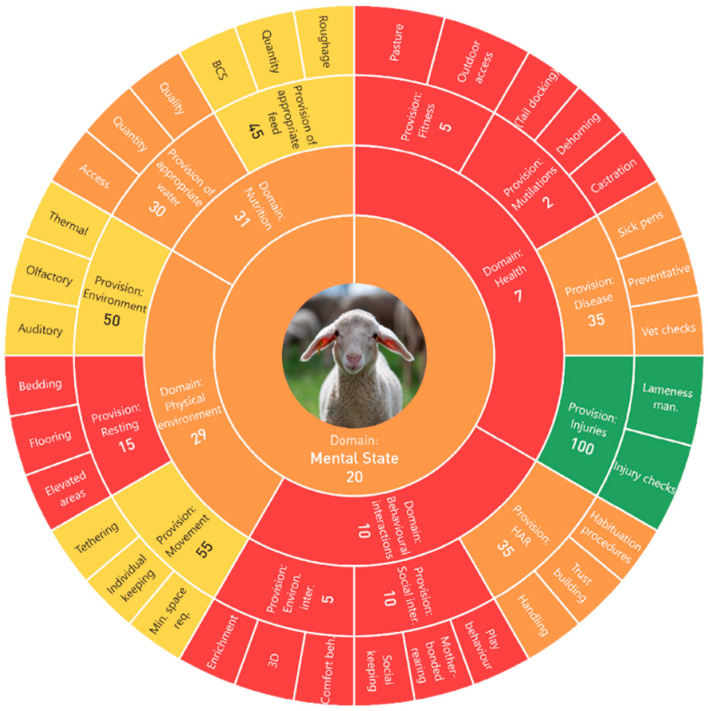
Standard 10-result chart.

**Table 1 animals-13-03696-t001:** Number of animals used in the textile industry, per material, along with the main location of production.

Animal-Derived Material	Number of Animals	Main Production Location
Down and feather	3,462,311,900	China
Leather	777,032,767	N/A
Sheep wool	633,000,000	Australia
Cashmere	33,680,000	China, Mongolia
Alpaca wool	4,367,816	Peru
Mohair	1,444,500	South Africa
Total	4,911,836,983	

**Table 2 animals-13-03696-t002:** Domains and provisions as per the Five Domains model.

Domain	Nutrition	Physical Environment	Health	Behavioural Interactions	Mental State
Provisions	Species-appropriate feed and water	Appropriate shelter and housing with comfortable living conditions	Prevention and treatment of diseases and injuries, also by ensuring proper fitness levels	Appropriate environment and interactions with other animals and humans	Safe and species-appropriate opportunities for experiencing pleasurable encounters in their lives

**Table 3 animals-13-03696-t003:** Provision mutilations, per species.

Animal	Mutilation	Scoring Influence
Alpaca	castration	method and use of pain relief
Goats, sheep, cattle	dehorning	method and use of pain relief
	tail docking	method and use of pain relief
	castration	method and use of pain relief
Sheep	mulesing	complete prohibition needed for allocation of points
Waterfowl	live plucking	complete prohibition needed for allocation of points
	force feeding	complete prohibition needed for allocation of points
	any form of flight restraint	complete prohibition needed for allocation of points

**Table 4 animals-13-03696-t004:** Animal welfare risk assessment—aggregation flowchart.

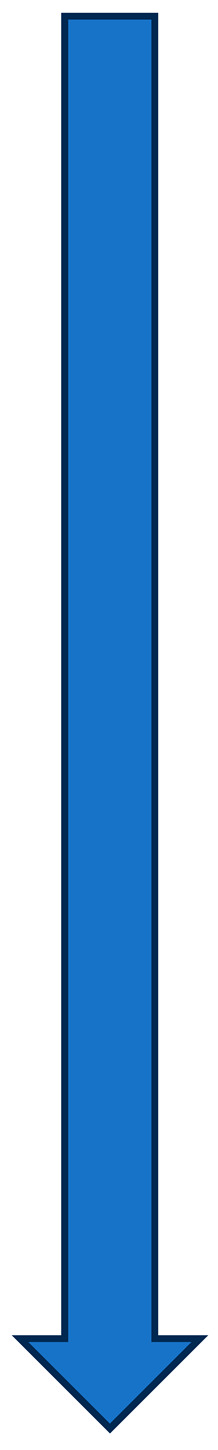	Single measure	Questions regarding the requirements, such as “Is the Body Condition Score of the animals regularly monitored?”
	Aggregation of single-measure scores into one score per provision with the use of a literature- and expert-informed decision tree
Provision	The provisions were defined with the use of the five domains of the animal welfare framework (appropriate feed, appropriate water, environmental comfort, resting comfort, ease of movement, absence of injuries, absence of disease, mutilations, fitness, environmental enrichment, social behaviour, human–animal relationship)
	Provision scores were aggregated with the use of a Choquet integral to obtain one score per domain
Domain	Four domains of animal welfare, namely nutrition, physical environment, health, and behavioural interactions
	Domain scores were aggregated using a median for a final mental state score
Mental state score	Final score, the mental state score

**Table 5 animals-13-03696-t005:** Animal welfare risk legend.

Score	Animal Welfare Risk
0–19	Probability for very poor animal welfare
20–39	Probability for poor animal welfare
40–59	Probability for acceptable animal welfare
60–79	Probability for good animal welfare
80–100	Probability for excellent animal welfare

**Table 6 animals-13-03696-t006:** Decision tree for the appropriate feed provision.

Single Measure Question		Single Measure Question		Single Measure Question		Provision Score
Ad libitum roughage	yes	Appropriate quantity of feed	yes	BCS management	yes	100
					no	65
			no	BCS management	yes	55
					no	35
	no	Appropriate quantity of feed	yes	BCS management	yes	45
					no	25
			no	BCS management	yes	35
					no	5

**Table 7 animals-13-03696-t007:** Individual anonymised textile standard results.

	Provision Scores		Domain Score	Provision Scores			Domain Score	Provision Scores				Domain Score	Provision Scores			Domain Score	
Nr.	Feed	Water	Nutrition	Envir.	Rest	Move.	Physical Environment	Injury	Disease	Mutil.	Fitness	Health	Enrich.	Social	HAR	Behavioural Interactions	Final Score
1.	45	55	46	50	45	55	48	100	100	0	35	19	65	100	35	49	47
2.	25	45	28	20	5	5	6	25	35	0	5	4	5	5	35	9	7.5
3.	5	30	8	20	5	5	6	25	15	0	5	3	5	5	35	9	7
4.	45	45	45	50	15	55	29	100	35	0	55	13	5	10	35	10	21
5.	25	30	26	50	5	5	6	40	35	0	5	5	5	10	5	6	6
6.	35	45	36	50	15	55	29	100	35	0	55	13	15	5	35	10	21
7.	45	100	53	50	5	30	16	40	35	0	5	5	5	10	35	10	13
8.	45	45	45	50	15	55	29	100	35	0	55	13	5	10	35	10	21
9.	45	30	31	5	5	15	6	100	35	0	55	13	5	65	35	19	16
10.	45	30	31	50	15	55	29	100	35	2	5	7	5	10	35	10	19.5
11.	45	30	31	20	5	55	15	100	35	17	5	14	5	10	35	10	14.5
12.	45	30	31	50	5	55	23	25	35	0	5	4	5	65	35	19	21
13.	5	5	5	5	5	5	5	5	35	0	5	3	5	5	35	9	5
14.	25	45	28	20	5	20	12	25	45	100	35	32	60	10	35	18	23
15.	25	15	15	50	5	20	11	25	15	0	5	3	5	5	35	9	10
16.	45	5	7	100	5	5	12	100	5	0	5	2	5	5	5	5	6
17.	45	5	7	100	45	55	51	100	35	17	55	25	15	20	35	18	21.5

## Data Availability

The data presented in this study are available on request from the corresponding author. The data are not publicly available due to this being the first assessment of its kind.

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
