# Peer review of "Assessing Animal Welfare Risk in Fibre-Producing Animals by Applying the Five Domains Framework"

_animals, 2023, doi:10.3390/ani13233696_

Round 1

Reviewer 1 Report

Comments and Suggestions for Authors

I would like to thank the authors of the paper for looking into an undoubtedly neglected sector, the textile one, which uses considerable numbers of animals and species. The results are really interesting. The study shows that, even within the tiny proportion of certified systems, a certification does not mean that the animals experience reasonable standards of well-being. This is an aspect that certainly warrants more scrutiny, more investigation and more work to improve industry standards also to ensure transparency towards consumers.

From a methodological perspective, I struggle a bit with the expert elicitation part as it determines the weight of the various components of the final welfare score. I would like to see a more detailed explanation of how the scores were obtained (see my specific comments below). 

I do hope that the paper can be published soon. 

L 38 Not that I am personally against using the terminology but I think that it will be difficult for an academic journal to accept wording such as "cruel" for practices that, unfortunately, are considered standard ones by the industry (and that are, also regrettably, mostly lawful). I would suggest replacing "cruel", "inhumane" and similar terminology with "painful", "ethically questionable", "hardly justifiable", etc. I think that this will increase your chances of a speedy publication :)

Here (38-39) for instance I would suggest "painful husbandry procedures and ethically questionable practices". I would recommend modulating the choice of wording accordingly throughout the article - meaning I will not comment on further instances. It's just my piece of advice. 

L 59 Perhaps "sum" is not the correct terminology here. The mental state resulting from all the experiences in the other domains is the result of their complex interactions and not just a matter of subtracting or adding (as summation would suggest). I would say something like "the effect/result of the various experiences on the mental state of the animal"....

L 61 See previous comment. It is not just a matter of numbers but also the intensity, frequency, and type of positive vs. negative experiences that determines the mental state of the animal. 

L 64-65 Consider adding sub-sections to introduce this part and the next one (L 83) as the shift in topic is quite abrupt.

Table 1, 3rd row: please check, there appears to be a problem with the text. Make sure the table includes a description and is referenced in the text. 

L 92 A reliable, science-based system?

L334-335 I think that, unless better explained, this expert elicitation that the authors evoke throughout the paper will be challenged. It is one of the most important elements of the paper because it determines to a large extent the relative importance of the various indicators and it is not described at all. I see that the authors use various references but there needs to be an explanation (in an Appendix, I would suggest) of the procedure and exact sources (was there one source, or multiple sources?). If the authors used multiple sources, how did the authors decide which ones to use? Or did they also consult experts themselves? 

Author Response

Thank you for your feedback. Please see the attachment.

Reviewer 2 Report

Comments and Suggestions for Authors

The paper, titled “Assessing Animal Welfare risk in fibre-producing animals by applying the Five Domains framework“ addresses an important and timely topic. I found the subject matter of the article fascinating and read the manuscript with great interest. The paper aligns well with the scope of the journal. However, I believe that in its current form, it has several shortcomings.

This paper addresses a pressing issue in the textile industry: the welfare of the approximately 5 billion farm animals involved in textile production. It aims to develop a risk assessment tool to evaluate various textile standards that promise improved animal welfare. The study analyzed 17 different standards and found that only one reached an acceptable level of risk for animal welfare, while the rest fell short. This research highlights the need for substantial improvements in animal welfare within the textile industry and raises questions about whether the current certifications meet these standards. The strength of the paper lies in its innovative approach to assessing animal welfare in the textile industry and its potential to drive positive change in this sector.

This paper is well-aligned with the scope of the journal. It addresses a critical issue related to animal welfare, which is relevant to the field of textiles and the treatment of farm animals in this industry. The research's focus on developing a risk assessment tool to evaluate various textile standards for their impact on animal welfare is within the scope of the journal, which covers topics related to animals in various industries and their welfare.

The main question addressed by the research is the development of a risk assessment tool to evaluate the impact of various textile industry standards on animal welfare, specifically concerning the treatment of farm animals involved in textile production.

The topic is both original and relevant in the field as it addresses the urgent need to assess the welfare of farm animals in the textile industry, where previous research and standards have often fallen short in providing adequate animal welfare protection.

The research adds significant value to the subject area compared to other published material by introducing a risk assessment tool specifically tailored to evaluate animal welfare in the textile industry. This tool is a novel approach to addressing animal welfare concerns in this context.

Regarding methodology, the authors might consider further details about the specific criteria used in the risk assessment tool. Providing more transparency about how the tool was developed and applied could strengthen the methodology section. Additionally, the authors should clarify how they determined the "acceptable" level of risk for animal welfare, as this is a crucial component of their assessment. It's also important to explain the selection criteria for the 17 textile standards assessed.

The conclusions appear consistent with the evidence and arguments presented. The study effectively demonstrates that most textile standards do not meet an acceptable level of animal welfare risk, highlighting the need for improvement in this industry.

The references seem appropriate and relevant to the research topic, providing a foundation for understanding the issues related to animal welfare in the textile industry.

I would like to point out that the paper does not sufficiently consider the variations in legislation and the treatment of animals worldwide. Animal welfare standards and practices differ significantly from one region to another, and this is a critical aspect that should be addressed in the paper. Ignoring this global variation may lead to misconceptions about the state of animal welfare practices within the textile industry.

Moreover, there are certain statements in the paper that I believe provide inaccurate information regarding the alleged cruel practices experienced by animals involved in the textile industry. It is essential to ensure that all claims made in academic publications are well-substantiated and based on accurate information to avoid the spread of misinformation and the potential damage to the reputation of industries.

I would like to request that the authors review and provide a more nuanced discussion on the differences in legislation and animal treatment practices worldwide, especially as they relate to the textile industry. Additionally, any claims made about cruel practices should be thoroughly researched, accurately presented, and supported by credible sources.

In this introduction, we aim to provide a comprehensive overview of various farming methods and their impacts on animal welfare. Several examples in the literature demonstrate the diverse approaches taken in different regions and production systems. Notable studies that shed light on these variations include:

10.3390/vetsci10090554 for calves

10.1016/j.rvsc.2023.03.008 for beef

10.3390/ani13050797 for dairy goats

10.3390/ani12141740 and 10.1186/s12917-022-03289-2 for horses

10.1080/1828051X.2020.1827990 for rabbits

10.3390/ani10122386 and 10.3390/ani10060945 for pigs

These studies serve as valuable references to illustrate the differences and complexities within the field of farming practices and their implications for animal welfare. We will draw upon these examples to provide context and a more nuanced understanding of the topic.

Author Response

(The authors gave the same response as above.)

Round 2

Reviewer 2 Report

Comments and Suggestions for Authors

Good job!